# Morphological and Molecular Identification of *Dactylogyrus gobiocypris* (Monogenea: Dactylogyridae) on Gills of a Model Fish, *Gobiocypris rarus* (Cypriniformes: Gobionidae)

**DOI:** 10.3390/pathogens12020206

**Published:** 2023-01-28

**Authors:** Jiangwen Cheng, Hong Zou, Ming Li, Jianwei Wang, Guitang Wang, Wenxiang Li

**Affiliations:** 1State Key Laboratory of Freshwater Ecology and Biotechnology and Key Laboratory of Aquaculture Disease Control, Ministry of Agriculture, Institute of Hydrobiology, Chinese Academy of Sciences, Wuhan 430072, China; 2College of Science, Tibet University, Lhasa 850000, China; 3University of Chinese Academy of Sciences, Beijing 100049, China

**Keywords:** dactylogyrid infection, rare minnow, *Gobiocypris rarus*, *Dactylogyrus gobiocypris*, histopathology

## Abstract

The rare minnow *Gobiocypris rarus* is an ideal model organism for toxicological research. *Dactylogyrus* species are usually found on the gills of this rare minnow in laboratory farming systems. Dactylogyrid infection may change the sensibility of fish to toxicants and affect toxicological evaluations. In the present study, dactylogyrid infection was investigated, and species of *Dactylogyrus* collected from rare minnows were determined. Based on the observed ‘*D. wunderi*’ type anchors, with a shorter outer root and elongated inner root, and accessory piece consisting of two parts, the dactylogyrids were identified as *D. gobiocypris*. A partial 18S-ITS1 rDNA sequence was firstly sequenced, and the highest sequence identity (86.7%) was to *D. cryptomeres*. Phylogenetic analysis revealed that *D. gobiocypris* formed a clade with *D. squameus*, *D. finitimus*, and *D. cryptomeres*, all of which have been recorded in the family Gobionidae. Histopathology analysis indicated that a heavy burden of *D. gobiocypris* caused necrosis of gill filaments. Inflammatory responses, such as tumefaction and hyperaemia, were also observed on gills with severe dactylogyrid infection. Supplementary morphological characteristics and 18S-ITS1 rDNA sequence provided basic data for identification of this parasite species.

## 1. Introduction

The rare minnow *Gobiocypris rarus* Ye et Fu, 1983 (Gobionidae) is a small gobionid fish endemic to China, mainly distributed at the edge of the west and northwest area of the Sichuan Basin [1,2]. This rare minnow possesses particular biological characteristics, such as high sensitivity to chemicals, small body size, short life cycle, and being easy to rear in laboratory, which make it an excellent model organism for ecotoxicology studies [3,4,5]. Since 1995, rare minnows have been widely used in acute and subchronic toxicity experiments on heavy metals [6,7], organics [8,9,10], and endocrine-disrupting chemicals [11,12,13]. 

The monogenean family Dactylogyridae Bychowsky, 1933 is one of the most species-rich groups of helminths, with more than 1000 species recognised worldwide [14]. Forty-one species of *Dactylogyrus* have been recorded from fishes in the family Gobionidae in China [15], and twenty-six species of *Dactylogyrus* are found on fishes in the family Gobionidae in Europe [16]. Species of *Dactylogyrus* were found on gills of rare minnows in a laboratory in China, and *D. gobiocypris* Yao, 1995 was described based on sclerotized parts of the anchor and copulatory complex [17]. No studies have reported *D. gobiocypris* since. 

Dactylogyrids can infect the gills of cypriniform fishes [18], causing serious hyperplasia of the gill filament epithelium, copious mucus, and eventually affecting respiratory function [19,20]. Fish heavily infected with dactylogyrids are also susceptible to bacterial infections [21,22,23]. 

The present study provides supplementary morphological characteristics, novel sequences of the 18S ribosomal RNA subunit and the first internal transcribed spacer region of rDNA (ITS1), and histopathological analysis of *D. gobiocypris* parasitizing *G. rarus* specimens reared in the laboratory in the Institute of Hydrobiology, Chinese Academy of Sciences, Wuhan City, Hubei province, China.

## 2. Materials and Methods

### 2.1. Parasite Collection

Samples of rare minnow were obtained from the laboratory in the Institute of Hydrobiology, Chinese Academy of Sciences in April 2022. Thirty fish (with total body length of 4.2 ± 0.9 cm) were randomly selected and anesthetized with 0.02% MS-222 (tricaine methanesulfonate) (Sigma, St Louis, MO, USA). Specimens of species of *Dactylogyrus* were then examined and collected using micro surgical needles under a stereoscopic microscope. Worms were rinsed several times with distilled water for further analyses.

### 2.2. Morphological Identification

A random subsample of dactylogyrids were mounted on a microscope slide and fixed in ammonium picrate glycerine (GAP) as whole mount following the procedure described by Ergens [24] and Malmberg [25] for morphological identification. GAP and Canada balsam were performed according to Ergens [24]. Additional specimens, with opisthaptors excised using a scalpel, were then individually subjected to proteolytic digestion according to the method described by Paladini et al. [26] and Tu et al. [22]. The tissue-free opisthaptoral sclerotized parts were mounted in GAP, and the excised body of each specimen was preserved in 95% ethanol for subsequent molecular analysis. Specimens were photographed using an optical microscope (Axioplan 2 imaging and Axiophot 2, Zeiss, Oberkochen, Germany). Measurements were taken according to Šimková et al. [27], and are given in micrometers (μm) unless otherwise stated. Identification of individual specimens was performed by comparing the morphology and measurements of anchors and the copulatory complex to the literature [17]. Five full worms, embedded in GAP and mounted on Canada balsam, were deposited as voucher specimens (accession nos. CJW-DG 202201-05) in the Museum of the Institute of Hydrobiology, Chinese Academy of Sciences, Wuhan City, Hubei province, China.

### 2.3. DNA extraction, Amplification and Sequencing

Genomic DNA was extracted from the excised bodies of 12 specimens using a Tissue Cell Genome Kit (TaKaRa, Beijing, China) according to the manufacturer’s instructions. The region of rDNA spanning the 3′ end of the 18S ribosomal RNA subunit, the entire ITS1 gene, and the 5′ end of the 5.8S ribosomal RNA subunit were targeted using primers S1 (5′-ATTCCGATAACGAACGAGACT-3′) and H7 (5′-GCTGCGTTCTTCATCGATACTCG-3′) [28,29]. PCR amplification was conducted using LA Taq polymerase (TaKaRa, Beijing, China) with the following profile: 5 min at 95 °C, 35 cycles of 1 min at 94 °C, 1 min at 55 °C, 1 min 30 s at 72 °C, and a final extension of 10 min at 72 °C. After purification, PCR products were cloned into the pGEM-T vector (Promega, Madison, USA), sequenced with the primers described above, produced by Sangon Biotech (Shanghai, China), and assembled manually with DNAStar’s SeqMan software (DNAStar, Madison, WI, USA).

### 2.4. Molecular Analyses

The obtained sequences of partial 18S rDNA, ITS1, and the flanking sequence of 5.8S rDNA were compared using BLAST in GenBank to assess similarity with other *Dactylogyrus* species. From the 12 specimens, 12 sequences of 18S-ITS1 rDNA were obtained to evaluate the intraspecific variation using BLAST. Sequences (Table 1) used for phylogenetic analyses were chosen from *Dactylogyrus* species from closely related hosts. *Thaparocleidus vistulensis* (Siwak, 1932), in the family Ancylodiscoididae, was used as the outgroup. Sequences were imported into PhyloSuite [30] and aligned with available 18S-ITS1 rDNA sequences of other *Dactylogyrus* spp. in GenBank using MAFFT 7.149 [31]. Ambiguously aligned fragments were trimmed using Gblocks [32] with the following parameter settings: minimum number of sequences for a conserved/flank position (6/6), maximum number of contiguous non-conserved positions (8), minimum length of a block (10), allowed gap positions (with half). Phylogenetic analyses based on the 18S-ITS1 sequences were performed using maximum likelihood (ML) and inference (BI) methods. TNe+G4 and K2P+G4 were chosen as the best-fit partition model for nucleotide evolution for ML and BI analyses, respectively using ModelFinder [33]. ML phylogenies were inferred using IQ-TREE [34], for 1000 standard bootstraps, as well as the Shimodaira–Hasegawa-like approximate likelihood-ratio test. BI phylogenies were inferred using MrBayes 3.2.6 [35], with two parallel runs (2,000,000 generations) in which the initial 25% of sampled data were discarded as burn-in. 

### 2.5. Histopathology Analysis

The first gill arch of each fish was collected and fixed in 10% neutral buffered formalin (Yeasen, Shanghai, China). The fixing solution was diluted to 4% after 4 to 24 h, washed for 24 h and dehydrated in graded ethanol. Gills were embedded in paraffin (Yeasen, Shanghai, China), sliced into 5 μm-thick sections, and stained with the Hematoxylin and Eosin Staining Kit (Yeasen, Shanghai, China) according to Molnár [36]. The slides were mounted on Canada balsam and examined under an optical microscope (Axioplan 2 imaging and Axiophot 2, Zeiss, Oberkochen, Germany).
pathogens-12-00206-t001_Table 1Table 1Species included in the phylogenetic analysis.Parasite SpeciesHost SpeciesLocalityGenBank IDRefs*Dactylogyrus cryptomeres**Gobio gobio*Morava River basin, Czech RepublicAJ564123[28]*Dactylogyrus finitimus**Romanogobio albipinatus*Morava River basin, Czech RepublicAJ564133[28]*Dactylogyrus squameus**Pseudorasbora parva*Morava River basin, Czech RepublicAJ564156[28]*Dactylogyrus gobiocypris**Gobiocypris rarus*Wuhan City, ChinaOP441417Present study*Dactylogyrus achmerowi**Cyprinus carpio*Morava River basin, Czech RepublicAJ564108[28]*Dactylogyrus extensus**Cyprinus carpio*Morava River basin, Czech RepublicAJ564129[28]*Dactylogyrus vastator**Carassius auratus*Liangzi Lake, ChinaKC876016[37]*Dactylogyrus intermedius**Carassius auratus*Liangzi Lake, ChinaKC876017[37]*Dactylogyrus lamellatus**Ctenopharyngodon idella*Morava River basin, Czech RepublicAJ564141[28]Outgroup



*Thaparocleidus vistulensis**Silurus glanis*Morava River basin, Czech RepublicAJ490165[38]


## 3. Results

### 3.1. Taxonomic Summary

#### *Dactylogyrus gobiocypris* Yao, 1995

Host: *Gobiocypris rarus* Ye et Fu, 1983 (Cypriniformes: Gobionidae).

Site of infection: gill filaments (Figure 1).

Locality: specimens collected from cultured rare minnow in the laboratory in the Institute of Hydrobiology, Chinese Academy of Sciences (30°54′74.1″ N, 114°35′’04.3″ E), Wuhan City, Hubei province, China.

Prevalence and mean abundance: 96.7% and 60.8 ± 84.5 (3–408), respectively.

Deposition of specimens: deposited in the Museum of the Institute of Hydrobiology (accession nos. CJW-DG 202201–05), Chinese Academy of Sciences, Wuhan City, Hubei province, China.

DNA reference sequences: a sequence (1042 bp) spanning the region of the 3′ end of the 18S ribosomal RNA subunit and ITS1 to the 5′ end of the 5.8S ribosomal RNA subunit was deposited in GenBank (OP441417).

Description: Based on 56 specimens fixed in GAP. Body length, 182 (118–248; n = 32); width, 45 (28–74; n = 32). Eye spots: two pairs. Pharynx diameter, 11 (7–17; n = 33). Copulatory complex: composed of penis and accessory piece, posterior to pharynx. Penis: tubular and well sclerotized; length, 12 (10–14; n = 30). Accessory piece: composed of two parts, one horseshoe-shaped and the other semicapsular; both intersect at the proximal part of the penis; length, 16 (13–17; n = 30). Vaginal armament: absent. Anchor: total length, 26 (24–30; n = 55); base length, 21 (18–24; n = 55); point length, 12 (10–13; n = 55); anchor inner root elongate length, 7 (6–9; n = 55); outer root length, 1 (1–2; n = 55). Ventral bar: rod-shaped, ends slightly enlarged, middle portion slightly convex posteriorly; total length, 4 (2–6; n = 56); median length, 2 (2–5; n = 56); width, 18 (15–21; n = 56). Dorsal bar: V-shaped, slightly extended; total length, 2 (2–4; n = 52); median length, 1 (1–2; n = 52); width, 16 (13–21; n = 52). Marginal hooks: seven pairs; total length, 17 (14–23; n = 55); shaft length, 12 (9–18; n = 55); sickle length, 5 (4–6; n = 55); filament loop length, 8 (7–9; n = 55) (Table 2) (Figure 2 and Figure 3). 

### 3.2. Morphological Characterization

*Dactylogyrus gobiocypris* Yao, 1995 was the only *Dactylogyrus* species described on the gills of this rare minnow in China [17]. More detailed morphometric measurements are provided herein, since new data on the morphology and phylogeny of *D. gobiocypris* were obtained in the present study. The measurements and shape of the sclerotized parts of the anchors of the specimens collected in the present study were almost identical to the original descriptions of *D. gobiocypris* by Yao [17]. However, the ventral bar was flatter and straighter, and the copulatory complex was shorter, than that of the *D. gobiocypris* described by Yao [17] (penis length 10–14 μm vs. 15–20 μm, and accessory piece 13–17 μm vs. 18–21, respectively). The morphometrical parameters of the sclerotized parts of dactylogyrids sometimes vary over seasons, temperature, and fixation and measurement procedure [29,40,41]. We used a substantial sample size in the present study, while the original descriptions by Yao were based on seven specimens. Thus, these discrepancies are judged to demonstrate intraspecific variation.

According to the studied morphological characteristics, *D. gobiocypris* most closely resembles *D. trullaeformis* in the shape of the anchors, with ‘*D. wunderi*’ type anchors, having a shorter outer root and elongated inner root. However, *D. gobiocypris* differs from *D. trullaeformis* [39] in: (1) the longer length (15–21 μm vs. 14–16 μm, respectively) and shape of the middle portion of ventral bar (slightly convex posteriorly), which is flatter and straighter in *D. trullaeformis*; (2) the accessory piece of *D. gobiocypris* consists of two parts, one of which is horseshoe-shape and the other semicapsular, while the accessory piece of *D. trullaeformis* consists of only one part and is shaped as a heterogeneous groove (Figure 4).

### 3.3. Molecular Analyses

Sequences (18S-ITS1) collected from the 12 specimens were identical; the length was 1042 bp. The BLAST search showed that *Dactylogyrus gobiocypris* displayed the highest sequence identity, 86.7%, to *D. cryptomeres*, which was collected from *Gobio gobio* (Linnaeus, 1758) (Cypriniformes: Gobionidae). The sequence of *D. gobiocypris* was then submitted in GenBank for the first time. 

Phylogenetic analyses, based on the BI and ML criteria of the 18S rDNA-ITS1 sequence, showed identical topology and only minor differences in statistical support values for some nodes (Figure 5). *Dactylogyrus gobiocypris* formed a clade with *D. squameus*, *D. finitimus*, and *D. cryptomeres*, all of which parasitize on the family Gobionidae. *D. lamellatus,* parasitic on *Ctenopharyngodon Idella* (Valenciennes, 1844) (Cypriniformes: Xenocyprididae), then formed a clade with those *Dactylogyrus* species above.

### 3.4. Histopathology Analysis 

The histopathological responses of the host to *D. gobiocypris* were investigated by serially sectioning the gills of naturally infected fishes. The gill lamellae of uninfected *G. rarus* were structurally intact, with consistent thickness at the base and end, uniform morphology, and visible gaps between gill lamellae (Figure 6 A,C). Histological examination showed that the infected gills were damaged, to some extent, by necrosis. Additionally, the infection caused hyperplasia of the respiratory epithelium between gill lamellae, with a tendency for adjacent gill filaments to fuse (Figure 6 B). Gill lamellae were affected by the anchors of *D. gobiocypris*, with a breakdown of cell integrity. Cell proliferation was also observed on the base of gill lamellae, which resulted in adhesion of adjacent gill lamellae (Figure 6D).

## 4. Discussion

Species of the genus *Dactylogyrus* are a group of monogenean gill parasites that are highly specific to freshwater fishes of the family Cyprinidae [28]. Basing on the measurements and shape of sclerotized parts of opisthaptor and copulatory complex, the dactylogyrids collected from gills of a rare minnow were identified as *D. gobiocypris*. To date, *D. gobiocypris* represents the only monogenean species reported infecting this rare minnow in China [17]. The present study provides additional measurements of sclerotized structures of the opisthaptor of this species, along with its molecular characterization and histopathological responses. 

In general, the taxonomy of dactylogyrid monogeneans depends on accurate descriptions of the size and shape of the sclerotized parts of the opisthaptor and reproductive organs [29]. The measurements and morphology of the sclerotized parts of the specimens collected in the present study were almost identical to those of *D. gobiocypris* provided by Yao [17]. Of the other species infecting closely related hosts in the Gobionidae family, *D. gobiocypris* most closely resembles *D. trullaeformis* in the shape of the anchors. However, there are distinct differences in the structure of the copulatory complex between the two species. The accessory piece of *D. gobiocypris* consists of two parts, one of which is horseshoe-shaped and the other semicapsular, whereas in *D. trullaeformis* it consists of only one part and is shaped as a heterogeneous groove. The 18S ribosomal RNA subunit and the internal transcribed spacer region (ITS1) are common molecular markers for identification of *Dactylogyrus* species [29,42]. The results of the BLAST search suggested the sequence of *D. gobiocypris* displayed the highest overall identity (86.7%) to *D. cryptomeres*, collected from *Gobio gobio*. The sequence of *D. gobiocypris* was obtained and submitted in GenBank for the first time. 

Phylogenetic trees (BI / ML) of *Dactylogyrus* species, constructed based on partial 18S-ITS1 rDNA sequences, are divided into two clades: (1) one clade includes dactylogyrids from *Cyprinus carpio* (Linnaeus, 1758) and *Carassius auratus* Linnaeus, 1758, both representatives of Cyprinidae; (2) the other clade includes parasite species of *C. idella* (Xenocyprididae) and Gobionidae. *Dactylogyrus gobiocypris* exhibited a relatively close phylogenetic relationship with *D. squameus*, *D. finitimus*, and *D. cryptomeres*, all of which parasitize fishes of the Gobionidae family. The molecular phylogeny shows a consistent pattern of relationships among *Dactylogyrus* species. This suggests that there is a high degree of host specificity among the *Dactylogyrus* species that parasitize Gobionidae fishes, which has been displayed in previous molecular phylogenetic studies [28,42,43]. 

*Dactylogyrus gobiocypris* was found on all individuals of *G. rarus* investigated, with a high abundance which reached 60.8±84.5 parasites per fish. Prevalence and mean abundance of *Dactylogyrus* infection in cultured rare minnow under laboratory conditions is higher than *Dactylogyrus* spp. in wild and farmed goldfish *Carassius auratus* [42,44]. The IHB rare minnow is a closed laboratory animal colony, the offspring of 50 wild *G. rarus* specimens collected in Hanyuan County of Sichuan Province in 2006 and bred using methods that prevent inbreeding [45]. The higher prevalence and mean abundance of *Dactylogyrus gobiocypris* infection may be related to declining genetic diversity and regular supplementation of the number of susceptible hosts. *Dactylogyrus gobiocypris* can be achieved by in vivo culture under laboratory conditions, and the host is singly infected with *D. gobiocypris*. The rare minnow (*Gobiocypris rarus*)–*D. gobiocypris* artificial infection system can be used as a new host–parasite laboratory model, which will provide support for further investigation. 

Observation of histopathological sections of gills of *G. rarus* infected with *D. gobiocypris* indicated that *D. gobiocypris* infection could lead to damage of gill lamellae, causing serious hyperplasia and fusion of the gill filament epithelium. These lesions may reduce the area of gas exchange, affect the respiratory function of gills, and even cause potential secondary infections leading to serious disease with adverse consequences [20,21,46]. In the present study, *G. rarus* infected with a high abundance of *D. gobiocypris* did not have obvious typical clinical symptoms or high mortalities. This lack of symptoms is perhaps caused by decreased parasite virulence or increased host tolerance with a long coevolutionary history. 

Parasitic infection may be capable of modifying the resistance of the host to other stressors [46,47,48]. The susceptibility to toxicants of *G. rarus* may be affected under the stress of high abundance of *D. gobiocypris*, thus interfering with the outcome of toxicological evaluation [49]. Fish hosts infected with parasites have been proven to be more sensitive to toxicants than uninfected conspecifics [50,51,52,53]. Most research to date on tolerance to chemicals and environmental pollutants appears to have overlooked the effects of parasites. Therefore, parasite infection in model organisms should be considered during aquatic toxicity testing and chemical safety assessment.

## Figures and Tables

**Figure 1 pathogens-12-00206-f001:**
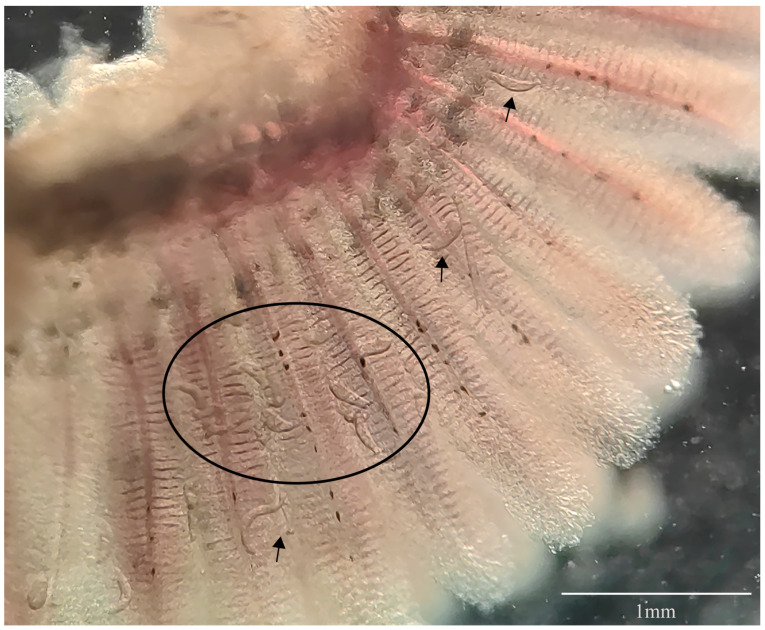
*Dactylogyrus gobiocypris* infection on gills of *Gobiocypris rarus*. Scale-bar: 1 mm.

**Figure 2 pathogens-12-00206-f002:**
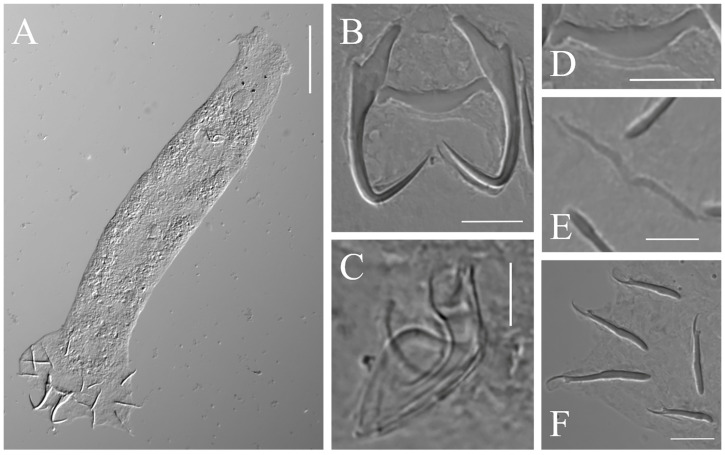
Light micrographs of *Dactylogyrus gobiocypris*: (**A**) whole parasite in ventral view; (**B**) opisthaptoral central hook complex; (**C**) copulatory complex (dorsal view); (**D**) ventral bar; (**E**) dorsal bar; (**F**) marginal hooks. Scale-bars: (**A**) 50 μm; (**B**,**D**,**F**) 10 μm; (**C**,**E**) 5 μm.

**Figure 3 pathogens-12-00206-f003:**
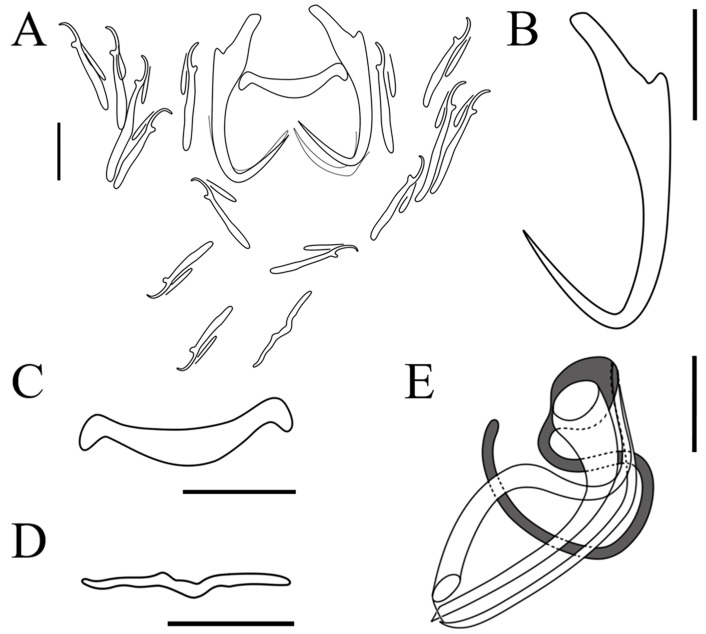
Line drawings of *Dactylogyrus gobiocypris*: (**A**) opisthaptor; (**B**) anchor; (**C**) ventral bar; (**D**) dorsal bar; (**E**) copulatory complex (dorsal view). Scale-bars: (**A**–**D**) 10 μm; (**E**) 5 μm.

**Figure 4 pathogens-12-00206-f004:**
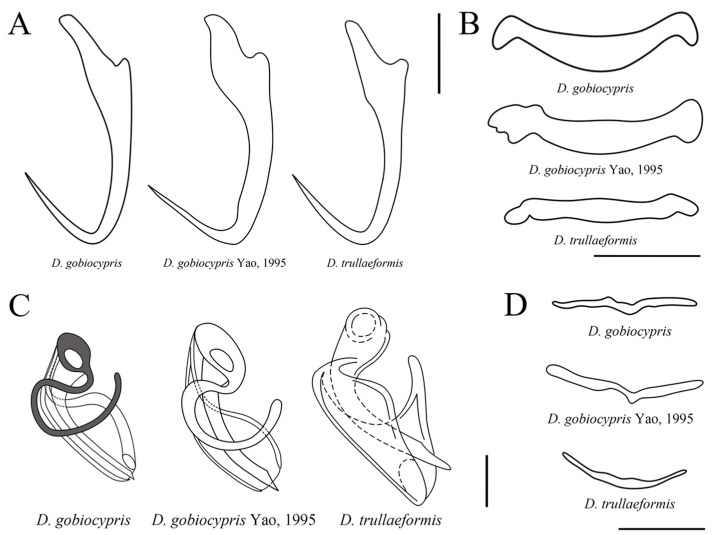
Comparisons of the opisthaptoral and copulatory sclerotized parts among *Dactylogyrus gobiocypris* in the present study, *D. gobiocypris* Yao, 1995 [17] and *D. trullaeformis* Gussev, 1955 [39]: (**A**) anchors; (**B**) ventral bars; (**C**) copulatory complex; (**D**) dorsal bars. Scale-bars: (**A**,**B**,**D**) 10 μm; (**C**) 5 μm.

**Figure 5 pathogens-12-00206-f005:**
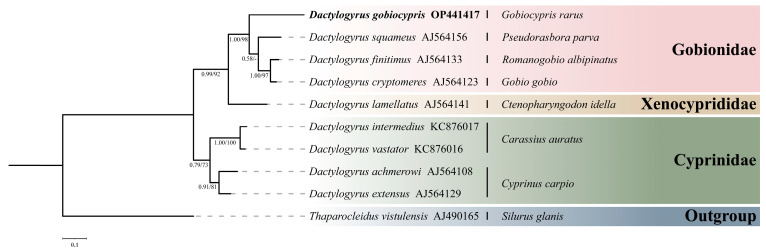
Phylogenetic analysis of *Dactylogyrus gobiocypris* estimated by Bayesian Inference, using 18S-ITS1 rDNA sequences of related species of *Dactylogyrus*. *Thaparocleidus vistulensis* was used as outgroup. Newly generated sequence is in bold. The higher taxa names to the right are for hosts. Posterior probabilities (BI) and bootstrap values (ML) are given below the nodes (posterior probabilities < 0.50 and bootstrap values < 50 are not shown).

**Figure 6 pathogens-12-00206-f006:**
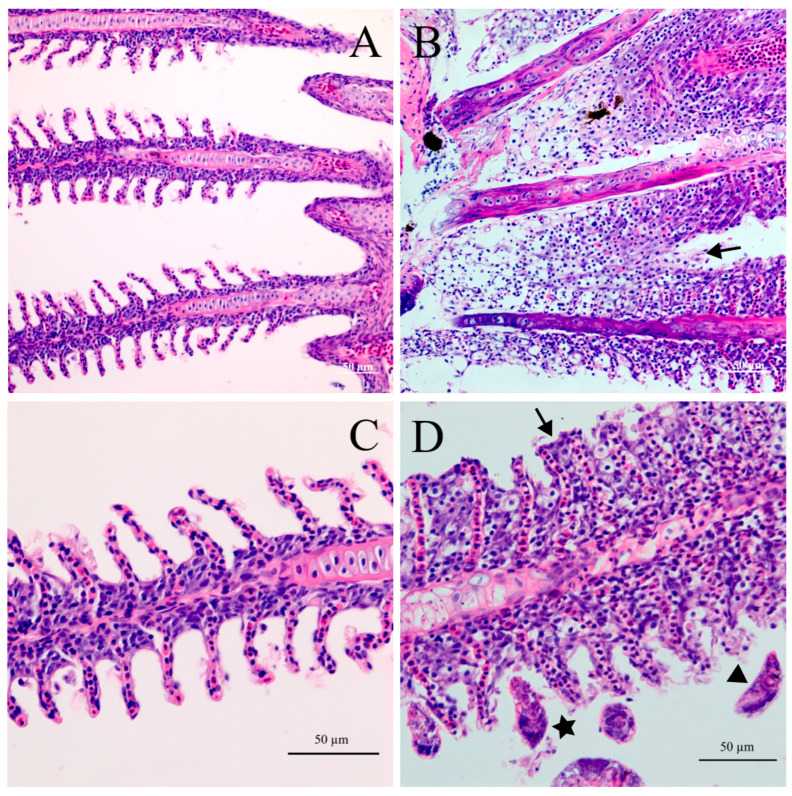
Comparison of histopathological sections of gills of *Gobiocypris rarus,* uninfected (**A**,**C**) and infected (**B**,**D**) with *Dactylogyrus gobiocypris*. The black arrow shows epithelial hyperplasia and the proliferating cells between adjacent gill lamellae. The black triangle indicates the end cells of the gill lamellae are damaged. The five-pointed star points to anchors of *D. gobiocypris* in gill tissues. Scale-bars: (**A**–**D**) 50 μm.

**Table 2 pathogens-12-00206-t002:** Morphometric parameters of *Dactylogyrus gobiocypris* in this study, *D. gobiocypris* Yao, 1995 [17] and *D. trullaeformis* Gussev, 1955 [39]. N, the number of *D. gobiocypris* specimens measured.

Source of Data	N	*Dactylogyrus gobiocypris*		*Dactylogyrus trullaeformis*
		*Gobiocypris rarus*	*G. rarus*	*Gnathopogon strigatus*
				*Squalidus chankaensis*
		this Study	Yao, 1995 [17] (n = 7)	Gussev, 1955 [39]
Body				
Total length	32	182.0 ± 34.6 (117.7–248.4)	102.5–113.0	150–300
Total width	32	44.8 ± 9.6 (27.8–74.1)	27.5–28.1	30–50
Pharynx diameter	33	11.2 ± 2.2 (7.2–16.8)	34.5	16–19
Anchor				
Total length	55	26.4 ± 1.6 (23.8–30.1)	26.5–30.0	27–30
Base length	55	21.1 ± 1.5 (18.2–24.3)		21–24
Outer root length	55	1.0 ± 0.2 (0.6–1.9)	1.5	1–2
Inner root length	55	7.2 ± 0.7 (5.9–9.0)	7.5–10.0	6–8
Point length	55	11.5 ± 0.8 (9.5–13.4)	12.5–13.0	11–13
Ventral bar				
Total length	56	3.6 ± 0.7 (2.4–5.7)		
Medium length	56	2.3 ± 0.6 (1.5–4.6)	1.8–3.4	1
Width	56	17.9 ± 1.4 (14.8–21.0)	17.5–20.0	14–16
Dorsal bar				
Total length	52	2.4 ± 0.4 (1.5–3.5)		
Medium length	52	1.0 ± 0.2 (0.5–1.5)	1.0	2
Width	52	15.8 ± 1.6 (13.0–20.6)	11.0–15.0	10–19
Marginal hook				
Total length	55	17.1 ± 1.5 (14.3–22.8)	16.5–25.0	15–23
Sickle length	55	5.0 ± 0.4 (4.0–6.3)		
Shaft length	55	12.1 ± 1.5 (9.3–17.7)		
Filament loop length	55	7.7 ± 0.5 (6.9–9.0)		
Copulatory complex				
Penis length	30	11.9 ± 1.2 (10.3–14.4)	15.0–20.0	16–21
Accessory piece	30	15.5 ± 1.1 (13.4–17.1)	17.5–21.3	17

## Data Availability

Not applicable.

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
