# Peer review of "Morphological and Molecular Identification of Dactylogyrus gobiocypris (Monogenea: Dactylogyridae) on Gills of a Model Fish, Gobiocypris rarus (Cypriniformes: Gobionidae)"

_pathogens, 2023, doi:10.3390/pathogens12020206_

Round 1

Reviewer 1 Report

This paper is staightforward and well written. I have only minor corrections and suggestions:

There is no abstract?

Title: FishBase currently recognises the Gobionidae as a family within the Cypriniformes.

Introduction

ln 14: According to FishBase it is a gobionid (vs a cyprinid).

ln 21: Need higher classification here I think, e.g. "(Dactylogyridae)".

ln 21-22: Is "richness" more appropriate than "diversity" here?

ln 22: Is "recognised worldwide" more appropriate than "found worldwide"?

ln 23: Replace "recorded on" with "recorded from".

ln 23: Insert "fishes" before "the subfamily". The worms were not recorded from the subfamily, rather, they were recorded from species of the subfamily.

ln 24: I think "and" is more appropriate than "while" here.

ln 24: Same problem as above, worms cannot be found on a subfamily, only species of that subfamily.

ln 25: Insert "a" before "laboratory"?

ln 26: Insert "the" before "anchor".

ln 27: This sentence is malformed. A species is reported, a study is not. I presume the authors mean that D. gobiocypris has not been reported since Yao, 1995, not that there have been no reports on studies of D. gobiocypris? I propose replace sentence with: "No studies have reported D. gobiocypris since."

ln 28: Is it true that dactylogyrids mainly infect cyprinid/cypriniform fishes? I thought they were known from a broad range of marine and freshwater fishes?

ln 28: The phrase "which cause" is inappropriate here, because "which" would seemingly refer to "cyprinid fishes" or perhaps "the gills of cyprinid fishes". Of course it is actually the dactylogyrids which should be the subject here. I think replace "which cause" with "causing".

ln 29: Replace "filaments" with "filament".

ln 29: I think replace "and" with a comma, before "copious mucus".

ln 30: Is "Fish" better than "Fishes" here?

ln 32: I think swap "morphological supplementary" to "supplementary morphological".

ln 32-33: I suggest replace "sequencing of" with "novel sequences for".

ln 33: "18S ribosomal DNA subunit" is not correct. The sequences generated are DNA, specifically 18S rDNA, but the subunit is the product and is rRNA, so "DNA subunit" does not make sense. Change "18S ribosomal DNA subunit" to either "18S ribosomal DNA subunit gene" or "18S ribosomal RNA subunit".

ln 33: Perhaps insert "first" before "internal", i.e. ITS1 vs ITS2. Also, specify that this is rDNA - internal transcribed spacer region of what?

Methods

ln 42: Specimens can relate only to a species, not a genus - one cannot collect specimens of a genus.

ln 44: Should "analysis" be "analyses", plural?

ln 46: Replace "was" with "were".

ln 74: Replace "analysis" with "analyses", plural - the authors report they used more than one analysis, i.e. ML and BI.

ln 75: "The obtained sequence" implies only a single sequence was generated, but in the section above, and the sentence immediately below, it is clear there were 12 replicates? Replace "sequence" with "sequences", plural.

ln 75: Replace rRNA with rDNA.

ln 77: I think "genotypes" is more appropriate than "haplotypes" here? But, in any case, there were 12 specimens and 12 sequences, but not 12 haplotypes/genotypes - there was only 1 - all 12 sequences were identical (according to results).

ln 77: The ITS1 is also rDNA, so why give as "18S rDNA-ITS1"? Why not "18S-ITS1 rDNA"?

ln 77: Above the region is given as 18S-ITS1-5.8S, here it is just 18S-ITS1. I suppose the 5.8S is only a short suffix, not enough to be of much use? This isn't made clear above. Maybe keep "partial" for 18S, but replace "partial" with something like "flanking sequence" with 5.8S?

ln 78: Probably do not need to specify 18S-ITS1 again here.

ln 79: "analyses", plural.

ln 79: How many taxa were included in the alignment?

ln 79: Interesting. Why not chose the most similar sequences from BLAST, or all comparable available sequences for Dactylogyrus spp? Why limit to those from similar fishes? And what are the bounds here - just from gobionids, other cypriniformes?

ln 80: Need an authority for this species. Is there a reason an outgroup from the Dactylogyridae (but not Dactylogyrus) could not be used?

ln 81: Why just ITS1 now - what happened to 18S?

ln 87: "maximum likelihood" and "inference" (but not "Bayesian") should be lowercase. These are not proper nouns, they are just types of analyses.

ln 87-89: I am confused as to why ModelFinder would suggest a different model for ML vs BI. I suspect, perhaps, what the authors are intending to report here is the models recommend under the AIC and BIC? In any case, these were the models suggested by ModelTest - but were these also the models used in the analyses? Is it possible to specify K2P+G4 using MrBayes?

ln 95: This sentence is confusing. Is the intended meaning: "From each fish, whether hosting dactylogyrids or not, the first gill arch was collected...". Or is the intended meaning: "For each fish hosting dactylogyrids, one infected gill arch and one uninfected gill arch were collected..." 

ln 99: Insert "mounted in Canada balsam and" before "examined".

ln 100-101: Delete "after were mounted in Canada balsam."

Results

ln 121: Include authority and family, perhaps order, for the host.

ln 124: Given the fish are lab reared, the coords are perhaps more precise than necessary, if needed at all?

ln 131: Elsewhere "ITS1" vs "ITS-1" here.

ln 133: End first sentence at "...GAP." Start next sentence using traditional telegraphic style, i.e. "Body 182 (117-248; n=32) long, 44 (27-74; n=32) wide."

ln 133: Removing trailing zero on "182".

ln 134: I suggest round numbers, fractions of micrometres not helpful.

ln 134: Use an en-dash, not a hyphen, for numerical ranges. Fix throughout.

ln 134: Start each sentence with the character: "Eye spots two pairs".

ln 135: Remove "is" and "the".

ln 136: Remove "located".

ln 136: Swap "Tubular penis" to "Penis tubular,".

ln 137: Remove "is".

lin 138: Replace "and" with a comma.

line 138: Remove "is", "of which", "the".

line 150: Remove "A".

line 156: "Penis" and "Accessory" should not be capitalised.

ln 157-159: The authors have a substantial sample size. What was the sample size for Yao? If small, might this explain minor discrepancy too?

ln 168: "analyses", plural.

ln 169: Should this be "identical"?

ln 171: Authority? Higher taxa?

ln 177: Why common name? What is "grass carp"? Common name is fine, but provide scientific name and higher taxa too.

ln 179: No, the response of D. gobiocypris was not investigated! The response of the host to D. gobiocypris was investigated.

ln 186: I suggest remove "which are attached to the end of them".

Discussion

ln 205: "fishes".

ln 219: Remove "above" - it is clear which two species are being referred to here.

ln 220: Replace "while" with "whereas in" and at "it" before "consisted".

ln 224: Remove trailing zero. Is the decimal precision necessary?

ln 225: Remove "which".

ln 227: Replace "partly" with "partial".

ln 229: Do these need authorities?

ln 232: Replace "on" with "fishes of".

ln 234: I think use present tense here.

ln 240: Insert "is" before "higher"?

ln 255-258: This sentence is not quite correct grammatically.

Table 1. Include the gene/marker name in the table caption. Make clear what the abbreviations "D." and "T." represent in the caption.

Table 2. Is it possible to give the N for Yao and Gussev?

Figure 5. Remove "Species name precedes the GenBank sequence ID" - that is obvious. Make clear that that higher taxa names to the right are for hosts, not for the taxa included in the analysis. Replace "18S rDNA and ITS1" with "18S-ITS1 rDNA", to make clear it is not a concatenated alignment, and because ITS1 is also rDNA.

Reviewer 2 Report

This manuscript describes morphological and molecular characteristics of the parasite Dactylogyrus gobiocypris together with pathological changes upon infection of gills of Gobiocypris rarus. The obtained 3’ end of 18S, the complete ITS1 and 5’ end of 5.8S seems to be the first and only D. gobiocypris sequence submitted to GenBank. The study adds additional morphological characteristics of the D. gobiocypris. The figures are appropriate and of good quality except the rather poor resolution of figure 5, at least in the PDF provided. The English of the manuscript is good.

Comments:

Please indicate manufacturer etc. of:

LA Taq polymerase

ammonium picrate glycerine (GAP)

Canada balsam

Tissue Cell Genome Kit

10% neutral buffered formalin

Paraffin

Hematoxylin

Eosin

 Do the word “indent” in line 169 mean that the sequence of the 12 specimens were identical?

Author Response

Comments and suggestions for Authors:

This manuscript describes morphological and molecular characteristics of the parasite Dactylogyrus gobiocypris together with pathological changes upon infection of gills of Gobiocypris rarus. The obtained 3’ end of 18S, the complete ITS1 and 5’ end of 5.8S seems to be the first and only D. gobiocypris sequence submitted to GenBank. The study adds additional morphological characteristics of the D. gobiocypris. The figures are appropriate and of good quality except the rather poor resolution of figure 5, at least in the PDF provided. The English of the manuscript is good.

Response: Thank you for the detailed evaluation. High quality image of the figure 5 was provided in the MS.

Comments:

Please indicate manufacturer etc. of:

LA Taq polymerase

ammonium picrate glycerine (GAP)

Canada balsam

Tissue Cell Genome Kit

10% neutral buffered formalin

Paraffin

Hematoxylin

Eosin

Response: Thank you for your reminder. Manufacture have been added to some of products.

Do the word “indent” in line 169 mean that the sequence of the 12 specimens were identical?

Response: Yes, sorry for our carelessness. We have changed the wrong formulation throughout the text.